# OpenMeta: A Comprehensive Multi-Task Benchmark for Metagenomics Understanding

## Abstract

Metagenomics is essential for exploring the vast diversity and intricate interactions of microbes that impact health, agriculture, and environmental sciences. Despite the surge of machine learning-based metagenomic models addressing these questions, evaluating their respective benefits is challenging due to the use of distinct, experimental datasets, partly contrived, and varying model performance across different tasks. To this end, we introduce OpenMeta, the first comprehensive benchmark tailored for metagenomic function prediction, which integrates diverse datasets ranging from 1,000 to 213,000 sequences and incorporates hierarchical data. We highlight the inadequacies of current genomic models and the superior performance of metagenomic pre-trained models for handling complex metagenomic data. Furthermore, we identify a critical research gap: the lack of unified models that process both sequence and hierarchical data. Addressing this could significantly advance metagenomic analyses. OpenMeta sets a new standard for metagenomic analysis, offering insights that could enhance the understanding and application of microbial ecology in biotechnology and environmental science.

## 1 Introduction

Metagenomics is a discipline that studies the genetic composition and functional dynamics of all microorganisms in environmental samples [41, 44]. By directly extracting the entire DNA from these microorganisms, metagenomics captures a broad spectrum of life forms, including viruses, viroids, and free DNA present in diverse habitats such as soil, seawater, and human microbiomes [53, 40]. Unlike traditional genomics, which focuses on sequencing DNA from single species in isolation [57, 74], metagenomics eliminates the need for isolating each organism, allowing research of uncultivable microorganisms [83, 48, 23]. Consequently, metagenomics unveils the vast diversity of microbial communities, enabling the interpretation of gene interactions and essential biological processes within ecosystems [3, 37, 39].

Deep learning techniques have significantly advanced metagenomics research, enabling more precise function prediction and complex relationship elucidation in metagenomics [1, 60, 45, 77, 22, 38](Sec. 2). However, the field lacks standardized benchmarks, making it difficult to evaluate the efficacy of various metagenomic models that often rely on distinct, artificially constructed datasets. Existing genomic benchmarks primarily focus on single-species genomic analysis. The GUE benchmark [84], built upon DNABERT2, encompasses multiple datasets ranging from humans to viruses and includes 7 binary classification tasks like promoter detection and transcription factor prediction. GenomicBenchmarks [49] address gene regulation and chromatin accessibility tasks. Nucleotide

Submitted to the 38th Conference on Neural Information Processing Systems (NeurIPS 2024) Track on Datasets and Benchmarks. Do not distribute.

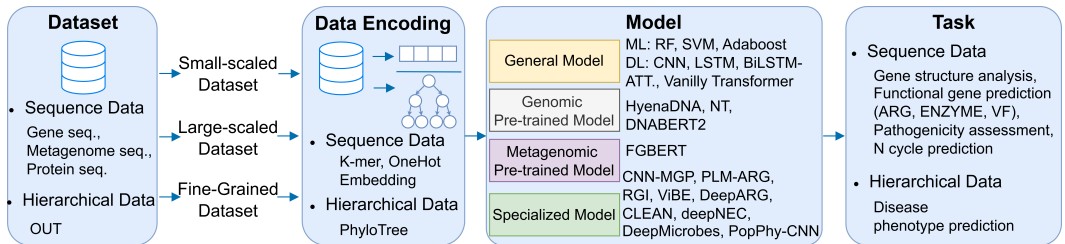

Figure 1: Framework of OpenMeta, including dataset preprocessing, data encoding, general and specialized models pre-trained or not, and target tasks.

Transformer (NT) [9], pre-trained on multi-species genomic data, concentrates on transcription factor binding and enhancer-promoter interactions within the human genome [32] (Sec. 3.1).

However, these methods often fail to address the unique challenges in metagenomics. *Firstly, metagenomic data is distinct* as it comprises genomic collections from multiple organisms sequenced simultaneously from numerous individuals, unlike genomic studies that focus on single organisms. *Secondly, metagenomic tasks are unique*, involving the prediction of complex interactions among various microorganisms, including bacteria and viruses, influenced by dynamic environments – a complexity that surpasses single-species genomic analysis. *Lastly, existing genomic models often perform poorly on metagenomic tasks*, as models trained on genomic data cannot be directly transferred to metagenomic data. Hence, there is an urgent need to develop a metagenomic reference to deepen our understanding of microbial ecology and provide a powerful tool for biotechnology and environmental science research (Sec. 3.2).

To this end, we develop OpenMeta, the **first** comprehensive benchmark specifically designed for metagenomics based on the FGBERT model [13]. OpenMeta integrates a wide range of tasks across genetic, functional, bacterial, and environmental levels and can handle datasets with sequences ranging from 1,000 to 213,000, as well as richer hierarchical data of phylogenetic tree structures, reflecting the diversity and complexity of metagenomic data. We compare metagenomic pre-trained models with genomic pre-trained models (Sec. 5.2), and although the latter provided new insights, their performance on metagenome data tended to decrease, highlighting the need for the development of metagenome pre-trained models. Our main contributions are as follows:

  i. We establish OpenMeta, the first comprehensive benchmark for metagenomic research encompassing 23 representative models. OpenMeta sets a new standard in the evaluation of metagenomic models using various collected standardized datasets and metrics across three dimensions: pre-trained vs. not pre-trained models, general vs. specialized models, and sequence data-based vs. hierarchical data-based models.

 ii. We conduct extensive experiments on various tasks ranging from metagenomic sequences to hierarchical data, covering small-scale, large-scale, and fine-grained scopes.

iii. Our findings lead us to reconsider the potential of metagenomic pre-trained models, advocating for architectures like FGBERT that better accommodate the diversity and complexity of metagenomic data.

 iv. We identify a significant research gap: the lack of a unified model capable of simultaneously processing sequence and hierarchical data. Addressing this could significantly advance comprehensive metagenomic analyses and represent a promising direction for future research.

## 2   Related Work

**Gene Representation Learning. For sequence metagenomic data,** while the K-mer method [17] efficiently captures characteristics of short sequences, it struggles with longer sequences due to its inherent limitations. Alternatively, one-hot encoding, despite its high-dimensional and sparse nature, restricts its utility for large-scale applications. In contrast, deep learning-based embeddings, such as those from Transformer models, enhance sequence representation by capturing contextual and global features, offering biologically meaningful insights. **For hierarchical metagenomic data,** constructing phylogenetic trees provides an effective framework for delineating hierarchical and evolutionary relationships among microbial taxa [2, 79]. This process begins with analyzing

microbial genomes through multiple sequence alignment, organizing them into a phylogenetic structure. Microbial taxa abundances are then mapped to corresponding nodes on the tree. By aggregating abundance values from child nodes to their respective parent nodes and transforming this phylogenetic tree into a matrix format [18, 58], the structure is adapted for input into CNN models for disease phenotype prediction.

**Metagenomic Methods.** Traditional alignment-based methods like MetaPhlAn5 [66] aim to match similarities between query sequences and known reference genomes and are common for taxonomic profiling. Advancements in deep learning have led to new methods like DeepVirFinder [61], which use CNNs for viral classifications with one-hot encoding. K-mer tokenization [17], employed in approaches like MDL4Microbiome [33], is a standard for DNA sequence characterization. Additionally, Virtifier [45] maps a nucleotide sequence using a codon dictionary combined with LSTM to predict viral genes. DeepMicrobes [35] employs a self-attention mechanism, while DeepTE [81] uses K-mer inputs with CNNs for element classification, and Genomic-nlp [46] applies word2vec for gene function analysis. MetaTransformer [77] uses K-mer embedding for species classification with Transformer. For pre-training models, LookingGlass [24] uses a three-layer LSTM model for functional prediction in short DNA reads. ViBE [22] employs a K-mer token-based BERT model pre-trained with Masked Language Modeling for virus identification.

**Genomic Benchmark.** To our knowledge, there is no benchmark in the field of metagenomics. Due to the scarcity of specialized benchmarks in metagenomics and the inherent similarities in data structure and content between genomic and metagenomic datasets, comparing the two allows us to leverage advancements in genomics to address metagenomics' unique needs. Existing genomic benchmarks, such as GUE, GenomicBenchmarks, and Nucleotide Transformer [9], each have their distinct focus but primarily address single-species genome analysis. GUE, as part of DNABERT2 [9], addresses challenges in genome tokenization and pre-training, covering multi-species datasets from humans to viruses and involving tasks like promoter detection and transcription factor prediction. GenomicBenchmarks, through HyenaDNA [49], focuses on improving long-sequence genome modeling and handling ultra-long sequences, including gene regulation and chromatin accessibility analysis. NT is pre-trained on multi-species genomic data, emphasizing transcription factor binding and enhancer-promoter interaction in the human genome.

# 3 Background

## 3.1 Difference Between Genomics and Metagenomics

The **Main** distinction between genomics and metagenomics lies in the **Number of organisms evaluated in a sample** in Tab. 1. Genomics focuses on the genome of a single organism, whereas metagenomics examines the collective genomes of different organisms within a sample [31, 63, 52]. **Sample type:** Genomics targets the complete genetic information of a single organism, typically from individual cells or

Table 1: Differences in Metagenomics and Genomics.

| Comparison Factor | Metagenomics | Genomics |
|---|---|---|
| **Main difference:** Organisms number | Many | One |
| Sample type | Genomes from many individuals within an environment | Individual organism's genetic makeup |
| Data complexity | Large, mixed data from multiple organisms mixed together | Relatively small, well-structured data from single organism |
| Sequencing depth | 3-100M Reads | 3-6M Reads |
| Sample types | Stool, Skin, Soil, Water | Cell Culture |
| Cost | Higher | Lower |

species, while metagenomics analyzes mixed DNA from multiple organisms in an environmental sample, allowing scientists to study unculturable microorganisms [48, 23]. **Data complexity:** Genomic data involves well-structured genetic information from a single organism, making it relatively simple. Metagenomic data includes genomes from various organisms, leading to large, mixed, and complex data. **Sequencing depth:** Metagenomic samples require significantly greater sequencing depth than single-genome sequencing. **Common sample types:** Metagenomic samples are derived from environments containing multiple genomes, such as soil, water, and human microbiomes, and are inherently more complex than genomic samples, which typically come from single-cell cultures [15].

**Cost of sequencing**: The extensive sequencing depth required for metagenomics generally results in higher costs than genomic sequencing.

## 3.2 Necessity of Developing Metagenomic Benchmarks

Tab. 2 provides a detailed comparison of downstream tasks for genomic and metagenomic pre-trained models. DNABERT2 [84] primarily engages in binary classification tasks such as promoter detection in human, mouse, and yeast species. Notably, **Transcription Factor Prediction** task recurs identically for both human and mouse, indicating consistent yet singular difficulty, which may reduce the overall challenge. HyenaDNA [49] is limited to regulatory elements classification tasks, reflecting a narrow scope. Additionally, **Demo and Dummy Datasets** are typically used for initial testing, lacking authenticity and practical value, reflecting the simplicity and limitations of its datasets. NT [9] covers 18 downstream tasks centered on splice site prediction and chromatin accessibility analysis.

While these genomic benchmarks perform well in single-species analyses, they often fail to capture the inherent complexities of multi-species interactions present in metagenomics. Specifically, the limitations include (1) **reliance on single-species data**, overlooking the complex interactions in metagenomics; (2) **lack of data diversity**, insufficient environmental diversity required for metagenomics; (3) **limited functional prediction**, focusing on sequence-based predictions without integrating crucial functional annotations; and (4) **inadequate model adaptability**, as models trained on single-species genomic data struggle to adapt to multi-species metagenomic data. In contrast, FGBERT [13], a metagenomic pre-trained model, aims to address interactions within different microbial communities. Its downstream tasks span multiple species, with a large number of classification categories, reflecting the diversity and complexity of metagenomic data. Therefore, incorporating FGBERT's multi-species metagenomic datasets into OpenMeta can enhance its ability to decipher complex microbial functions. For detailed analysis, please refer to Appendix B.

Table 2: Comparison of pre-trained models for genome and metagenome.

| | Model | Category | Task | #Class |
|---|---|---|---|---|
| Genome | DNABERT2 | Human | **Transcription Factor Pred.** | **2** |
| | | | Promoter Detection | 2 |
| | | | Splice Site Detection | 3 |
| | | Mouse | **Transcription Factor Pred.** | **2** |
| | | Virus | Covid Variant Class. | 9 |
| | HyenaDNA | Human | Regulatory | 2 |
| | | **Demo Dataset** | Elements | **2** |
| | | **Dummy Dataset** | Classification | **2** |
| | NT | Yeast | Epigenetic marks Pred. | 10 |
| | | - | Splice site Pred. | 2 |
| | | - | Chromatin Profiles Pred. | 919 |
| Meta. | FGBERT | Mixed Multi-Species | Gene Structure Pred. | 1379 |
| | | | ARG Pred. on Gene Family | 269 |
| | | | Virulence Factor Pred. | 15 |
| | | | Pathogenic Genes Pred. | 110 |

# 4 OpenMeta

## 4.1 Supported Methods

OpenMeta supports 23 methods for comprehensive analysis and performance evaluation across various tasks and data types, detailed in Tab. 3 with their respective publication years. To present them systematically, we categorize them along three dimensions: General vs. Specialized Models, Pre-trained vs. Not Pre-Trained Models, and Sequence data-based vs. Hierarchical data-based Models, acknowledging that some methods may span multiple categories.

**General vs. Specialized Models:** General models include widely used machine learning and deep learning models, such as SVM [69], Random Forest (RF) [64], CNN, LSTM [25], and Transformer [76], providing robust foundations for various tasks. Moreover, models like DNABERT2 [84] will be discussed in detail as pre-trained models in the next part. Conversely, specialized models are designed for specific metagenomic tasks such as functional gene prediction (PLM-ARG [78], DeepARG [5], RGI [4], ViBE [22], CLEAN [82]), and prototype prediction (PopPhy-CNN [59]).

**Pre-trained vs. Not Pre-trained Models:** OpenMeta includes DNABERT2 [84], HyenaDNA [49], and NT [9] genomic pre-trained models, alongside FGBERT [13] metagenomic pre-trained model. Although ViBE [22] and PLM-ARG [78] are not pre-trained from scratch, they use BERT model [11] and protein language model [36], respectively, to enhance their functional prediction capabilities.

Table 3: Categorizations of all supported metagenomic methods in our OpenMeta. RF denotes Random Forest, and VT represents Vanilla Transformer.

| Model | Pre-Trained | Not Pre-Trained | General | Specialized | Sequence-based | Hierarchical-based | Multi-Classification | Binary-Classification | Year | Description |
|---|---|---|---|---|---|---|---|---|---|---|
| RF | | ✓ | ✓ | | ✓ | | ✓ | | | Machine Learning |
| SVM | | ✓ | ✓ | | ✓ | | ✓ | | | Machine Learning |
| AdaBoost | | ✓ | ✓ | | ✓ | | ✓ | | | Machine Learning |
| LSTM | | ✓ | ✓ | | ✓ | | ✓ | | | Deep Learning |
| BiLSTM | | ✓ | ✓ | | ✓ | | ✓ | | | Deep Learning |
| VT [76] | | ✓ | ✓ | | ✓ | | ✓ | | | Deep Learning |
| FGBERT [13] | ✓ | | ✓ | | ✓ | | ✓ | | 2024 | Metagenomic pre-trained model for functional prediction. |
| HyenaDNA [49] | ✓ | | ✓ | | ✓ | | ✓ | | 2023 | Genomic pre-trained model trained on multi-species genomes. |
| NT [9] | ✓ | | ✓ | | ✓ | | ✓ | | 2023 | Genomic pre-trained model trained over human reference genome. |
| DNABERT2 [84] | ✓ | | ✓ | | ✓ | | ✓ | | 2023 | Genomic pre-trained model trained on diverse human genomes. |
| CNN-MGP [1] | | ✓ | | ✓ | ✓ | | | ✓ | 2019 | Gene prediction using CNN network. |
| PlasGUN [16] | | ✓ | | ✓ | ✓ | | | ✓ | 2020 | Gene prediction tool using multiple CNN network. |
| PLM-ARG [78] | ✓ | | | ✓ | ✓ | | ✓ | | 2023 | ARG identification framework using a pretrained protein language model. |
| DeepARG [5] | | ✓ | | ✓ | ✓ | | ✓ | | 2018 | ARG prediction software by alignment and metagenomic sequences. |
| RGI [4] | | ✓ | | ✓ | ✓ | | ✓ | | 2023 | ARG prediction tools for annotating genes from scratch. |
| DeepVirFinder [61] | | ✓ | | ✓ | ✓ | | | ✓ | 2020 | Viral sequences prediction with reference and alignment-free CNNs. |
| ViBE [22] | ✓ | | | ✓ | ✓ | | ✓ | | 2022 | Eukaryotic viruses identification with hierarchical BERT model. |
| ViraMiner [71] | | ✓ | | ✓ | ✓ | | | ✓ | 2019 | Viral genomes identification in human samples. |
| DeepVF [80] | | ✓ | | ✓ | ✓ | | | ✓ | 2021 | Viral factor identification with hybrid framework using stacking strategy. |
| HyperVR [27] | | ✓ | | ✓ | ✓ | | | ✓ | 2023 | Viral factors and mixing of ARG simultaneous prediction. |
| CLEAN [82] | | ✓ | | ✓ | ✓ | | ✓ | | 2023 | Enzyme function prediction using contrastive learning. |
| DeepMicrobes [35] | | ✓ | | ✓ | ✓ | | | ✓ | 2020 | Taxonomic classification for metagenomics with self-attention model. |
| PopPhy-CNN [59] | | ✓ | | ✓ | | ✓ | ✓ | | 2020 | Host Phenotypes prediction by systematic tree embedded CNN network. |

**Sequence Data-based vs. Hierarchical Data-based Models:** This work further integrates models trained on hierarchical data, such as PopPhy-CNN [59], which leverages the phylogenetic tree structures of microbial communities to enhance the understanding of microbial interactions, contrasting with sequence-based models essential for raw genetic data analysis without considering the hierarchical relational structure among microbial taxa.

## 4.2 Supported Tasks

Table 4: Detailed information of metagenomic sequence datasets in OpenMeta.

| Type | Dataset | Category | #Seq. | #Cate. | Seqs/Cate Range (Min-Max) | Avg. Len. | Description |
|---|---|---|---|---|---|---|---|
| Small-Scale Classification | E-K12 [65] | Gene-pair Cls. | 4,315 | 1,379 | 1-106 | 510.96 | Tasks involving smaller datasets focusing on high accuracy in narrow contexts. |
| | CARD-A [28] | Gene-wise Cls. | 1,966 | 269 | 1-229 | 1088.1 | |
| | CARD-D [28] | | 1,966 | 37 | 1-513 | 1088.1 | |
| | CARD-R [28] | | 1,966 | 7 | 1-1263 | 1088.1 | |
| | PATRIC [19] | | 5,000 | 110 | 1-1081 | 307.82 | |
| Large-Scale Classification | ENZYME [6] | Gene-wise Cls. | 5,761 | 7 | 288-2055 | 426.76 | Tasks requiring handling of large data volumes, broad pattern extraction. |
| | VFDB [7] | | 8,945 | 15 | 5-1683 | 415.47 | |
| | NCycDB [75] | | 219,089 | 69 | 1-20548 | 347.03 | |
| Fine-Grained Classification | NCRD-N [42] | Gene-wise Cls. | 104,363 | 1912 | 1-18370 | 407.44 | Focused on detailed differentiation within closely related categories. |
| | NCRD-F [42] | | 104,363 | 420 | 2-35364 | 407.44 | |
| | NCRD-C [42] | | 104,363 | 29 | 1-14159 | 407.44 | |
| | NCRD-R [42] | | 104,363 | 10 | 166-38073 | 409.79 | |

Table 5: Detailed information of metagenomic hierarchical datasets in OpenMeta.

| Type | Dataset | Category | #Seq. | K | P | C | O | F | G | S | Description |
|---|---|---|---|---|---|---|---|---|---|---|---|
| Hierarchical Classification | Cirrhosis [56] | Metagenome-wise Cls. | 542 | 3 | 15 | 27 | 40 | 76 | 186 | 531 | Detailed profiling of microbial diversity linked to cirrhosis. |
| | T2D [30] | | 606 | 3 | 17 | 29 | 48 | 94 | 216 | 587 | Detailed analysis for Type 2 Diabetes-associated microbiota. |

We conduct extensive experiments across various **multi-classification tasks**, including gene structure analysis, functional gene prediction, pathogenicity assessment, nitrogen cycle prediction, and disease phenotype prediction, utilizing diverse datasets that range from metagenomic sequences to hierarchical data We provide detailed descriptions of the following 14 datasets in Tab. 4 and 5, covering small-scale, large-scale, and fine-grained scopes. Seqs/Cate Range provides the range of sequences in each category, from minimum to maximum. Details can be found in Appendix C.

**(1) Small-Scale Classification: Gene Operon Prediction Task.** This task aims to identify transcription factor binding sites with the strongest correlation with operon regulation in the gene regulatory network [8, 14, 51] This gene-pair classification utilizes E-K12 dataset [65], consisting of 4,315 operons, each detailed with operon names, descriptions, and gene components. **Antibiotic Resistance Genes (ARGs) Prediction Task.** Accurate identification of ARGs is essential for understanding the relationship between the microbiome and disease, as pathogenic microorganisms threaten public health by exacerbating ARGs to invade the host [50]. This gene-wise classification uses CARD dataset [28], categorizing genes by 269 AMR Gene Families (CARD-A), 37 Drug Classes

Table 7: Enzyme function Prediction on ENZYME.

| Method | ENZYME |
|---|---|
| RF (3-mer) | 33.6 |
| SVM (3-mer) | 31.3 |
| AdaBoost (3-mer) | 31.4 |
| LSTM (w2v) | 42.8 |
| LSTM (one-hot) | 34.1 |
| BiLSTM (w2v) | 38.7 |
| BiLSTM (one-hot) | 31.6 |
| BiLSTM-Att. (w2v) | 36.9 |
| BiLSTM-Att. (one-hot) | 43.6 |
| VT | 68.2 |
| HyenaDNA | 79.6 |
| NT | 74.1 |
| DNABert2 | 85.4 |
| FGBERT | 99.1 |
| CLEAN | 92.3 |

Table 8: Virus factor Prediction on VFDB.

| Method | VFDB |
|---|---|
| RF (3-mer) | 22.4 |
| SVM (3-mer) | 28 |
| AdaBoost (3-mer) | 27.3 |
| LSTM (w2v) | 36.7 |
| LSTM (one-hot) | 32.9 |
| BiLSTM (w2v) | 46.1 |
| BiLSTM (one-hot) | 31.3 |
| BiLSTM-Att. (w2v) | 37.7 |
| BiLSTM-Att. (one-hot) | 36.7 |
| VT | 58 |
| HyenaDNA | 61.1 |
| NT | 58.3 |
| DNABert2 | 58.2 |
| FGBERT | 75.7 |
| ViBE | 50.9 |

Table 9: N Cycling Prediction on NcycDB.

| Method | NCycDB |
|---|---|
| RF (3-mer) | 67 |
| SVM (3-mer) | 66.9 |
| AdaBoost (3-mer) | 68.8 |
| LSTM (w2v) | 71.9 |
| LSTM (one-hot) | 65 |
| BiLSTM (w2v) | 66.9 |
| BiLSTM (one-hot) | 82 |
| BiLSTM-Att. (w2v) | 69 |
| BiLSTM-Att. (one-hot) | 67.3 |
| VT | 84.5 |
| HyenaDNA | 92.4 |
| NT | 75.1 |
| DNABert2 | 88.6 |
| FGBERT | 99.5 |

(CARD-D), and 7 Resistance Mechanisms (CARD-R). **Pathogens Prediction Task.** This task assesses the pathogenic potential of pathogens to cope with the public health risks [29]. We use PATRIC core dataset [19], which has 5000 pathogenic bacterial sequences across 110 classes.

**(2) Large-Scale Classification: Enzymes Prediction Task.** Enzymes are important catalysts in living cells that produce essential molecules needed by living organisms through chemical reactions [73]. ENZYME dataset [6] contains 5,761 enzyme sequences, which are grouped into 7 classes according to their corresponding EC numbers. **Virulence Factors (VFs) Prediction Task.** Viruses are common in both humans and different habitats, and they are always changing. Therefore, accurately identifying VFs is extremely important for understanding the relationship between the microbiome and disease. VFDB dataset [7] for virulence factors prediction contains 8,945 VF sequences across 15 categories, detailing structural features, functions, and mechanisms of major bacterial pathogens. **Nitrogen (N) Cycling Process Prediction Task.** The N cycle is a collection of important biogeochemical pathways in the Earth's ecosystems, and quantitatively studying the functional genes related to the N cycle [20]. NCycDB dataset [75] contains 68 genes (sub)families and covers 8 N cycle processes with 219,089 representative sequences, each involving a specific gene family.

**(3) Fine-Grained Classification: ARG Prediction.** Targets a more precise and detailed prediction of antibiotic resistance properties, which helps in exploring ARG characteristics comprehensively and detecting potential resistant mechanisms [34]. NCRD dataset [42] is dedicated to the fine-grained categorization of microbial resistance genes, differentiating in detail between 420 Gene Families, 1,912 specific Gene Names, 30 major Resistance, and 10 different Mechanisms.

**(4) Hierarchical-Data Classification: Disease Prediction Task.** Predicting host phenotypes and identifying relevant markers are pivotal for unraveling the complexities of host-microbiome interactions [72, 55], and the impact of such interactions on disease [26, 43, 47] can be explored using the phylogenetic structure and relative abundance of microbial taxa [21]. Cirrhosis dataset [56] comprises 232 data cases on microbiome liver disease. Type 2 Diabetes (T2D) dataset [30] comprises 440 data cases on glucose metabolism disorders.

### 4.3 Evaluation Metrics

For multi-classification tasks, we use the Macro F1-score (M.F1) as the primary metric to accommodate the inherent class imbalance present within datasets. For the Fine-grained Benchmark, Accuracy, Precision, Recall, and False Negative Rate (FNR) are incorporated. FNR is particularly critical for ARG prediction scenarios where the consequences of overlooking true positives are severe, necessitating nuanced assessments of the model's ability to identify them reliably.

Table 6: Comparison of ARG prediction methods on CARD. (− means inability to predict specific category).

| Method | ARG Prediction | | |
|---|---|---|---|
| | CARD-A | CARD-D | CARD-R |
| RF (3-mer) | 22.4 | 36.1 | 47.8 |
| SVM (3-mer) | 27.6 | 33.6 | 43.3 |
| AdaBoost (3-mer) | 36.9 | 36.4 | 36.2 |
| LSTM (w2v) | 47.1 | 37.5 | 47.5 |
| LSTM (one-hot) | 46.2 | 39.1 | 41.5 |
| BiLSTM (w2v) | 43.3 | 35.5 | 36.3 |
| BiLSTM (one-hot) | 47.4 | 38.9 | 58.9 |
| BiLSTM-Att. (w2v) | 31.9 | 43.5 | 35.1 |
| BiLSTM-Att. (one-hot) | 46.7 | 31.2 | 41.6 |
| VT | 57.1 | 49.8 | 55.7 |
| HyenaDNA | 50.9 | 53.6 | 66.2 |
| NT | 58.5 | 56.2 | 68 |
| DNABERT2 | 65.2 | 51.5 | 61.2 |
| FGBERT | 78.6 | 57.4 | 69.4 |
| DeepARG | - | 52.2 | 65.3 |
| PLM-ARG | - | - | 68.1 |
| RGI | - | - | - |

# 5 Results and Insights

## 5.1 Results

**Small-Scale Benchmarks.** Appendix Tab. A5 and A6 show M.F1 for gene operon and pathogen prediction on two small-scale datasets, E-K12 and PATRIC (sequence length less than 5000). General models (RF to VT) test K-mer (K=3), one-hot, and w2v data encoding methods, and BiLSTM (one-hot) and LSTM (w2v) performed best. FGBERT achieves superior results, far exceeding other methods, highlighting the necessity of metagenomic pre-training. Models like CNN-MGP [1], PlasGUN [16], and DeepMicrobes [35], designed for binary classification, are unsuitable for our multi-classification benchmark. Tab. 6 evaluates various models on three ARG prediction datasets: CARD-A, CARD-D, and CARD-R. In General models, the word2vec data encoding method performs better. FGBERT outperforms HyenaDNA, NT, and DNABERT2. This suggests that genomic models may not be sufficient to cope with the complexity of metagenomic data, which involves multiple microbial interactions and environments. In Specialized models, DeepARG [5] performs well on CARD-D and CARD-R but unable to predict CARD-A. Since RGI [4] itself is based on CARD dataset, it is not included.

**Large-Scale Benchmarks.** Tab. 7, 8and 9 summarize M.F1 across three large-scale datasets (sequence length more than 5000): EN-ZYME, VFDB, and NCycDB, the analysis reflects a similar trend. FGBERT demonstrates exceptional efficacy across all datasets, significantly outperforming specialized models such as ViBE [22] and CLEAN [82]. Conversely, models designed for binary classification, such as ViraMiner [71], deepVF [80], HyperVR [27] and DeepVirFinder [61] do not align well with our benchmark's requirements. However, HyperVR's innovative approach to predicting VF and ENZYME concurrently has inspired potential developments in OpenMeta for simultaneous

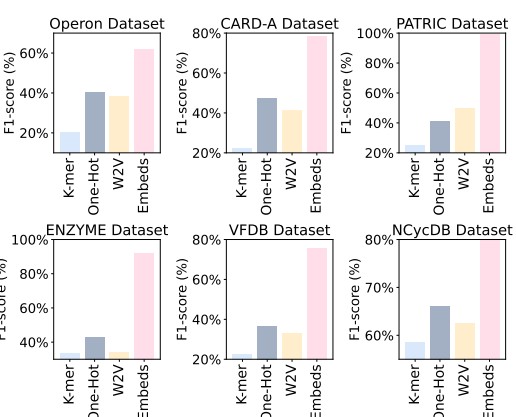

Figure 2: Comparison of different data encoding methods across tasks.

multi-task predictions. Fig. 2 compares different data encoding methods. Details can be found in Appendix E. From K-mer to language model representation, model performance gradually improves, indicating that capturing contextual information in the sequence is important.

**Fine-Grained Benchmarks.** The fine-grained NCRD dataset provides a more rigorous test for ARG prediction tasks. As depicted in Fig. 3, FGBERT performs well in all resistance categories, identifying 30 antibiotic classes. In comparison, DeepARG identified 17, and RGI identified 22, with the undetected classes shown in light gray. Moreover, the 4 methods have high accuracy in beta-lactam, aminoglycoside, and multidrug categories. Furthermore, Tab. 11 shows that FGBERT and RGI cover all NCRD classes, while DeepARG is limited to specific classes, likely due to limitations in its training data. PLM-ARG has a high false negative rate, as shown in Tab. 10, indicating its limitations in application.

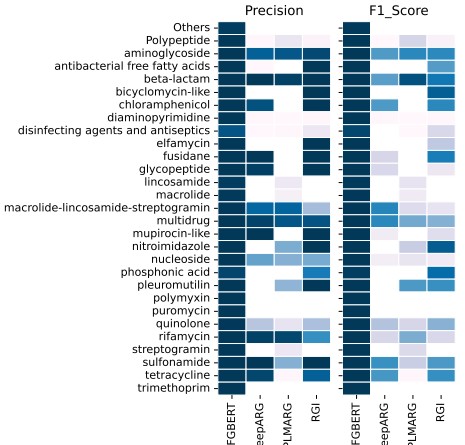

Figure 3: Comparison of different methods on each ARG category.

**Hierarchical-Data Benchmark.** In the host phenotype prediction task, we use microbial taxa's relative abundance values as input. Since abundance

Table 10: Comparison of M.F1 for ARG Prediction on different categories of NCRD Datasets. ($-$ indicates inability to predict specific categories. FNR is the false negative rate, lower is better).

| Method | NCRD-Gene Family (420) | | | | | NCRD-Gene Name (1900) | | | | | NCRD-Resistance (30) | | | | | NCRD-Mechanisms (11) | | | | |
|---|---|---|---|---|---|---|---|---|---|---|---|---|---|---|---|---|---|---|---|---|
| | Acc. | Pre. | Re. | F1 | FNR | Acc. | Pre. | Re. | F1 | FNR | Acc. | Pre. | Re. | F1 | FNR | Acc. | Pre. | Re. | F1 | FNR |
| DeepARG | - | - | - | - | - | 0.63 | 1.00 | 0.43 | 0.60 | 0.57 | 0.97 | 0.99 | 0.51 | 0.65 | 0.42 | - | - | - | - | - |
| RGI | 0.56 | 0.97 | 0.36 | 0.50 | 0.61 | 0.50 | 1.00 | 0.26 | 0.42 | 0.74 | 0.56 | 1.00 | 0.36 | 0.52 | 0.64 | 0.61 | 0.62 | 0.59 | 0.57 | 0.51 |
| PLM-ARG | - | - | - | - | - | - | - | - | - | - | 0.96 | 0.87 | 0.74 | 0.88 | 12.38 | - | - | - | - | - |
| FGBERT | 0.94 | 0.96 | 0.94 | 0.93 | 0.03 | 0.93 | 0.93 | 0.92 | 0.93 | 0.05 | 0.99 | 0.99 | 0.99 | 0.99 | 0.31 | 0.99 | 0.99 | 0.98 | 0.99 | 0.01 |

Table 14: Comparative Analysis of Pre-training Strategies in Genomic and Metagenomic Models.

| Model | Pre-training Dataset | Token | Network Architecture | Application Tasks | Benchmark |
|---|---|---|---|---|---|
| DNABERT2 | Human and multi-species genome | BPE | Advanced BERT with MLM+ ALiBi | Promoter detection, Transcription factor, binding site prediction | 28 datasets on GUE benchmark |
| HyenaDNA | Human reference genome | 6-mer | Simple stack of Hyena operators with NTP | Gene regulation prediction, chromatin accessibility analysis | 8 datasets on GenomicBenchmarks +18 prediction tasks on NT |
| NT | Human reference genome | 6-mer | Encoder-only Transformer + RoPE | Transcription factor binding, enhancer-promoter interaction prediction | 18 prediction tasks |
| FGBERT | Multi-species metagenome | Protein-based genomic representation | Advanced BERT with contrastive learning | Metagenomic sequences and functions analysis | 14 datasets on Metagenomic benchmark |

279 data alone cannot reveal the hierarchical structure among species and introduces data redundancy, we
280 adopt a phylogenetic tree-based modeling approach to process abundance data [12], effectively reduc-
281 ing redundancy and retaining species information. After constructing a phylogenetic tree through
282 multiple sequence alignment, abundance values are filtered and assigned to the tree's nodes, and the
283 values of child nodes are summed to their parent nodes. Finally, the phylogenetic tree is converted
284 into a matrix format for analysis. Tab. 12 shows that the specialized model PopPhy outperforms the
285 general models on Cirrhosis dataset. LSTM and Transformer models are not tested because they
286 are mainly applicable to sequence data and have difficulty capturing the hierarchical structure and
287 phylogenetic relationships between species. At present, no model can process both metagenomic
288 sequence and hierarchical phylogenetic tree data, indicating a key direction for future research.

## 5.2 Observations and Insights

290 **(A) Metagenomic Pre-trained Models vs. Genomic**
291 **Pre-trained Models:** Tab. 14 compares genomic
292 and metagenomic pre-trained models, including pre-
293 training datasets, token embedding methods, network
294 architectures, application tasks, and benchmarks. In
295 terms of **Pre-Training Datasets**, DNABERT2 [84]
296 utilizes human and multi-species genomes for its
297 foundational model pre-training, covering a vast
298 dataset of 27.5 billion nucleotide bases from the Hu-
299 man reference genome [32] and 135 species genomes
300 across seven categories. HyenaDNA [49], on the
301 other hand, is pre-trained solely on a single hu-
302 man reference genome. NT [9] pre-trains on three
303 datasets: the human reference genome [32], 3,202
304 diverse human genomes, and 850 genomes from sev-
305 eral species. In contrast, FGBERT [13] employs
306 MGnify database [62], comprising 2,973,257,435
307 metagenomic sequences from various microbial com-
308 munities. For **Token Embedding**, DNABERT2 ap-
309 plies Byte Pair Encoding (BPE) [67], while both Hye-
310 naDNA and NT use 6-mer tokenization. FGBERT

Table 11: Comparisons of different ARG pre-diction methods on NCRD.

| Method | Categories of NCRD Dataset | | | |
|---|---|---|---|---|
| | Gene Family | Gene Name | Resistance | Mechanisms |
| DeepARG | | ✓ | ✓ | ✓ |
| RGI | ✓ | ✓ | ✓ | ✓ |
| PLM-ARG | | | ✓ | |
| FGBERT | ✓ | ✓ | ✓ | ✓ |

Table 12: Performance of General vs. Specialized Models on Cirrhosis dataset.

| Method | M.F1 | AUC | MCC | Pre. | Re. |
|---|---|---|---|---|---|
| RF | 0.79 | 0.93 | 0.61 | 0.88 | 0.87 |
| SVM | 0.77 | 0.89 | 0.57 | 0.85 | 0.84 |
| AdaBoost | 0.70 | 0.71 | 0.43 | 0.72 | 0.71 |
| CNN | 0.84 | 0.89 | 0.68 | 0.81 | 0.80 |
| PopPhy | 0.81 | 0.90 | 0.61 | 0.83 | 0.82 |

Table 13: Performance of General vs. Specialized Models on T2d dataset.

| Method | M.F1 | AUC | MCC | Pre. | Re. |
|---|---|---|---|---|---|
| RF | 0.66 | 0.72 | 0.33 | 0.67 | 0.67 |
| SVM | 0.61 | 0.63 | 0.23 | 0.61 | 0.61 |
| AdaBoost | 0.70 | 0.70 | 0.42 | 0.70 | 0.70 |
| CNN | 0.59 | 0.65 | 0.19 | 0.60 | 0.59 |
| PopPhy | 0.58 | 0.64 | 0.18 | 0.59 | 0.58 |

311 utilizes a unique protein-based genomic representation tailored for metagenomic sequences. Re-
312 garding **Network Architecture**, both DNABERT2 and FGBERT adopt BERT-like structures [11];
313 DNABERT2 enhances its predecessor by replacing learned positional embeddings with Attention with
314 Linear Biases (ALiBi) [54] to eliminate input length limitations and incorporates Flash Attention [10]

to boost computational efficiency. FGBERT introduces contrastive learning to strengthen the intricate relationships between metagenomic sequences and functions, proposing two pre-training tasks to enhance co-representation learning of metagenomic gene sequences and functions. HyenaDNA employs a simple stack of Hyena operators for next token prediction, while NT uses an encoder-only Transformer architecture with Rotary Positional Embeddings (RoPE) [70] to enable reasoning over longer sequences during training.

**(B) Sequence Data vs. Hierarchical Data: Why Use Hierarchical Data?** Hierarchical data introduce an additional dimension by providing interrelationships and evolutionary context among microbial communities, enriching metagenomic research. Unlike traditional abundance data, hierarchical data offer not only quantitative information but also capture the complex hierarchical relationships between microbes, which is crucial for exploring host-microbe interactions [68].

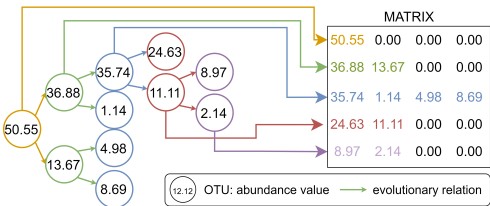

Figure 4: Phylogenetic Tree Representation of Microbial Communities for Hierarchical Data.

**Why Use Phylogenetic Tree Structures for Hierarchical Data?** Tree structures naturally represent the hierarchical and phylogenetic relationships among microbial taxa. Each node represents a microbial taxon, and the connections between nodes reflect their evolutionary relationships. This helps to reveal the evolutionary links between different microbial taxa, integrating complex biological information (such as abundance and hierarchical data) into a unified data structure. By accumulating the abundance values from child nodes to their parent nodes and converting the phylogenetic tree into a matrix format, each row represents a level in the tree, and columns represent different microbes or attributes. As shown in Fig. 4, this matrix-based representation effectively combines abundance and hierarchical information. This approach is particularly useful in disease prediction tasks, such as studies on Cirrhosis and T2D, demonstrating how understanding the hierarchical structure of microbial communities can elucidate the complexity of host-microbe interactions. This hierarchical method provides powerful tools for the precise identification and functional analysis of disease-related microbial communities. Our benchmark framework underscores the importance and benefits of using hierarchical data to enhance the accuracy and depth of metagenomic analysis.

# 6 Conclusion and Future Work

**Conclusion.** In this paper, we introduce OpenMeta, the first comprehensive benchmark tailored for metagenomic function prediction. This benchmark standardizes the evaluation process across various metagenomic tasks and facilitates the design of metagenomic models through a unified approach. Our extensive analysis includes comparisons between pre-trained and not pre-trained models, general versus specialized models, and sequence data-based versus hierarchical data-based models. Inspired by OpenMeta, we emphasize the necessity of pre-trained metagenomic models in this field and advocate for the community's engagement with metagenomic models trained on hierarchical data such as phylogenetic trees. This approach can profoundly enhance our understanding of the complex relationships and interactions within microbial communities.

**Limitations.** It is crucial to note that OpenMeta primarily serves as an evaluative tool that aggregates and assesses a wide array of multi-class datasets, including both sequence and hierarchical data. While this benchmark significantly contributes to the field, it does not involve the development of new models but focuses on the assessment of existing methodologies. This limitation underscores the necessity for further research and development in creating comprehensive models that can process both sequence and hierarchical inputs simultaneously.

**Future Work.** We identify a significant gap in the current landscape: the absence of a unified metagenomic model capable of simultaneously processing sequence and hierarchical data from phylogenetic trees. Addressing this gap represents a promising direction for future work and could significantly advance our holistic understanding of metagenomics.

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

# Supplement Material

## A  Social Impacts

Metagenomics offers significant benefits in fields like medicine and environmental science, yet it also poses dual-use concerns. For instance, technologies designed to minimize disease can theoretically be repurposed for harmful uses, such as biological weapons. The advancement of metagenomic benchmarks could inadvertently facilitate such misuse. Additionally, while these technologies can accelerate experimental processes, the need for traditional wet lab experimentation remains crucial. Thus, developing robust, clinically validated benchmarks is essential for integrating metagenomic methods into medical practice responsibly. This approach will ensure technological advances support health and environmental management without replacing foundational experimental techniques.

## B  Necessity of Developing Metagenomic Benchmarks

Table A1: Comparison of Downstream Tasks for Genomic and Metagenomic Pre-trained Models. The repetition of Transcription Factor Prediction tasks in the human and mouse categories suggests that it may not present significant challenges. Demo and Dummy datasets are usually artificially generated.

| Model | Category | Dataset | Task | #Seq. | #Class |
|---|---|---|---|---|---|
| DNABERT2 | Human |  | Core Promoter Detection (3) | 4,904 | 2 |
|  |  |  | **Transcription Factor Pred. (5)** | 32,378 | **2** |
|  |  |  | Promoter Detection (3) | 4,904 | 2 |
|  |  |  | Splice Site Detection (1) | 36,496 | 3 |
|  | Mouse |  | **Transcription Factor Pred. (5)** | 6,478 | **2** |
|  | Yeast |  | Epigenetic Marks Pred. (10) | 11,971 | 2 |
|  | Virus |  | Covid Variant Class. (1) | 77,669 | 9 |
| HyenaDNA | Human | Enhancers Cohn | Regulatory Elements Class. | 27,791 | 2 |
|  |  | Enhancers Ensembl |  | 154,842 | 2 |
|  |  | Regulatory |  | 289,061 | 3 |
|  |  | Nontata Promoters |  | 36,131 | 2 |
|  |  | OCR Ensembl |  | 174,756 | 2 |
|  | **Demo** | **Coding vs Intergenomic** |  | 100,000 | **2** |
|  |  | **Human vs Worm** |  | 100,000 | **2** |
|  | **Dummy** | **Mouse Enhancers** |  | 1,210 | **2** |
| NT | Yeast |  | Epigenetic marks Pred. (10) | 25,953 | 10 |
|  | - |  | Promoter sequence Pred. (3) | 59,194 | 2 |
|  | - |  | Enhancer sequence Pred. (2) | 14,968 | 3 |
|  | - |  | Splice site Pred. (3) | 19,775 | 2 |
|  | - |  | Chromatin Profiles Prediction (1) | - | 919 |
|  | - |  | Enhancer Activity (1) | 14,968 | 50 |
| FGBERT | Mul. Spe. | E. coil K12 | Gene Structure Pred. | 4,315 | 1379 |
|  | Mul. Spe. | CARD-A | ARG Pred. on AMR Gene Family | 1,966 | 269 |
|  | Mul. Spe. | CARD-D | ARG Pred. on Drug Class | 1,966 | 37 |
|  | Mul. Spe. | CARD-R | ARG Pred. on Resistance Mechanism | 1,966 | 7 |
|  | Mul. Spe. | VFDB | Virulence Factor Pred. | 8,945 | 15 |
|  | Mul. Spe. | ENZYME | Enzyme Function Pred. | 5,761 | 7 |
|  | Mul. Spe. | PATRIC | Pathogenic Genes Pred. | 5,000 | 110 |
|  | Mul. Spe. | NCycDB | N Cycling Genes Pred. | 213,501 | 68 |

In our analysis of applications and benchmarks, Table A1 provides a detailed comparison of downstream tasks for genomic and metagenomic pre-trained models, highlighting datasets, tasks, sequence numbers, and class counts. DNABERT2 [84] primarily engages in binary classification tasks such as promoter and splice site detection in human, mouse, and yeast datasets. The numbers in parentheses indicate the number of independent sub-datasets for each task, and the sequence count reflects the size of the first sub-dataset. Notably, the **Transcription Factor Prediction** task recurs identically for both human and mouse species, suggesting a uniform difficulty level and potentially reducing the challenge due to its repetition across similar species. HyenaDNA [49]'s downstream tasks are divided into two parts: GenomicBenchmarks, which includes 8 regulatory element classification datasets with sequence lengths ranging from 200 to 500, and NT's 18 prediction tasks. Beyond human

Table A2: The detailed information of supported datasets in OpenMeta with source link.

| Model | Pre-Trained | Not Pre-Trained | General | Specialized | Sequence-based | Structure-based | Year | Link | Description |
|---|---|---|---|---|---|---|---|---|---|
| SVM | | 51 | 51 | | 51 | | | | Machine Learning |
| RF | | 51 | 51 | | 51 | | | | Machine Learning |
| AdaBoost | | 51 | 51 | | 51 | | | | Machine Learning |
| CNN | | 51 | 51 | | 51 | | | | Deep Learning |
| LSTM | | 51 | 51 | | 51 | | | | Deep Learning |
| Vanilla Transformer | | 51 | 51 | | 51 | | | | Deep Learning |
| FGBERT | 51 | | 51 | | 51 | | 2024 | | Metagenomic pre-trained model for functional prediction. |
| DNABERT2 | 51 | | 51 | | 51 | | 2023 | https://github.com/MAGICS-LAB/DNABERT_2 | Genomic pre-trained model trained on multi-species genomes. |
| HyenaDNA | 51 | | 51 | | 51 | | 2023 | https://github.com/HazyResearch/hyena-dna | Genomic pre-trained model trained over human reference genome. |
| Nucleotide Transformer | 51 | | 51 | | 51 | | 2023 | https://github.com/instadeepai/nucleotide-transformer | Genomic pre-trained model trained on diverse human genomes. |
| CNN-MGP | | 51 | | 51 | 51 | | 2019 | https://github.com/rachidelfermi/cnn-mgp | Gene prediction using CNN network. |
| PlasGUN | | 51 | | 51 | 51 | | 2020 | https://github.com/zhenchengfang/PlasGUN | Gene prediction tool using multiple CNN network. |
| PLM-ARG | 51 | | | 51 | 51 | | 2023 | https://github.com/Junwu302/PLM-ARG | ARG identification framework using a pretrained protein language model. |
| DeepARG | | 51 | | 51 | 51 | | 2018 | https://github.com/gaarangoa/deeparg | ARG prediction software by alignment and metagenomic sequences. |
| RGI | | 51 | | 51 | 51 | | 2023 | https://github.com/arpcard/rgi | ARG prediction tools for annotating genes from scratch. |
| DeepVirFinder | | 51 | | 51 | 51 | | 2020 | https://github.com/jessieren/DeepVirFinder | Viral sequences prediction with reference and alignment-free CNNs. |
| ViBE | 51 | | | 51 | 51 | | 2022 | https://github.com/DMnBI/ViBE | Eukaryotic viruses identification with hierarchical BERT model. |
| ViraMiner | | 51 | | 51 | 51 | | 2019 | https://github.com/NeuroCSUT/ViraMiner | Viral genomes identification in human samples. |
| DeepVF | | 51 | | 51 | 51 | | 2021 | http://deepvf.erc.monash.edu/ | Viral factor identification with hybrid framework using stacking strategy. |
| HyperVR | | 51 | | 51 | 51 | | 2023 | https://github.com/jiboyalab/HyperVR | Viral factors and mixing of ARG simultaneous prediction. |
| CLEAN | | 51 | | 51 | 51 | | 2023 | https://github.com/tttianhao/CLEAN | Enzyme function prediction using contrastive learning. |
| DeepMicrobes | | 51 | | 51 | 51 | | 2020 | https://github.com/MicrobeLab/DeepMicrobes | Taxonomic classification for metagenomics with self-attention model. |
| PopPhy-CNN | | 51 | | 51 | | 51 | 2020 | https://github.com/YDaiLab/PopPhy-CNN | Host Phenotypes prediction by systematic tree embedded CNN network. |

datasets, HyenaDNA incorporates **Demo and Dummy** datasets, typically used for initial testing and validation, though they may lack the data authenticity and application value of specifically collected datasets. Furthermore, NT [9] covers 18 downstream tasks, primarily centered on transcription factor binding, promoter prediction, and chromatin accessibility analysis, emphasizing its close relationship with gene regulation mechanisms. While these genomic benchmarks perform well in single-species analyses, they often fail to capture the inherent complexities of multi-species interactions present in metagenomics. Specifically, the limitations of genomic benchmarks include (1) **reliance on single-species data**, which overlooks the complex interactions in metagenomics; (2) **lack of data diversity**, as their datasets are typically structured and uniform, lacking the environmental diversity required for metagenomic studies; (3) **limited functional prediction**, focusing on sequence-based predictions without integrating crucial functional annotations; and (4) **inadequate model adaptability**, as models trained on single-species genomic data struggle to adapt to multi-species metagenomic data.

In contrast, FGBERT [13], as a metagenomic pre-trained model, aims to address interactions within different microbial communities and predict functions across various environments, covering diverse tasks such as gene structure analysis, functional gene prediction, pathogenicity assessment, and nitrogen cycle prediction. These tasks span gene, functional, bacterial, and environmental levels, with input sizes ranging from 1,000 to 213,000 sequences, reflecting the diversity and complexity of metagenomic data. Therefore, incorporating FGBERT's multi-species genomic datasets into our OpenMeta benchmark not only substantiates its proficiency in deciphering complex microbial functions but also provides a solid framework for comparing and evaluating different models' performance in practical applications. This approach enhances our understanding and utilization of metagenomic pre-trained models in biotechnology and environmental science.

## C Datasets

We provide CSV files containing the categories and quantities of each dataset in the .zip fils.

We provide detailed descriptions of the 12 open-source datasets as shown in Appendix Table A2.

Appendix Table A3 shows detailed statistics for all sequence datasets in OpenMeta. The 'Num. Seqs.' column indicates the total number of sequences in the data set, and the 'Num. Cates' column shows the number of different categories in the data set. The 'Seqs/Cate Range' column provides the range of sequence numbers in each category, from smallest to largest. The 'Avg. Len.' column indicates the average length of the sequences. The 'Source' column describes the source of the data. The 'Task Type' column indicates the type of task for which the data set was used.

Appendix Table A4 shows detailed statistics for all hierarchical datasets in OpenMeta. The 'Hierarchical Taxonomic Levels' column means the Taxonomic Distribution in Metagenomic Datasets for

Table A3: Statistical analysis of all sequence datasets.

| Dataset | Num. Seqs. | Num. Cates | Seqs/Cate Range (Min-Max) | Avg. Len. | Source | Task Type |
|---|---|---|---|---|---|---|
| E-K12 | 4312 | 1379 | 1-106 | 510.96 | Public Database | Multi-Classification |
| CARD-A AMR Gene Family | 1966 | 269 | 1-229 | 1088.1 | Public Database | Multi-Classification |
| CARD-D Drug Class | 1966 | 37 | 1-513 | 1088.1 | Public Database | Multi-Classification |
| CARD-R Resistance Mechanism | 1966 | 7 | 1-1263 | 1088.1 | Public Database | Multi-Classification |
| PATRIC Pathogenic Genes? | 5000 | 110 | 1-1081 | 307.82 | Public Database | Multi-Classification |
| ENZYME | 5761 | 7 | 288-2055 | 426.76 | Public Database | Multi-Classification |
| VFDB | 8945 | 15 | 5-1683 | 415.47 | Public Database | Multi-Classification |
| NCycDB Nitrogen Cycling Genes | 219089 | 69 | 1-20548 | 347.03 | Public Database | Multi-Classification |
| NCRD-N Gene Name | 104363 | 1912 | 1-18370 | 407.44 | Public Database | Multi-Classification |
| NCRD-F Gene Family | 104363 | 420 | 2-35364 | 407.44 | Public Database | Multi-Classification |
| NCRD-C Categories | 104363 | 29 | 1-14159 | 407.44 | Public Database | Multi-Classification |
| NCRD-R Resistance Mechanism | 104363 | 10 | 166-38073 | 409.79 | Public Database | Multi-Classification |

Table A4: Statistical analysis of all hierarchical datasets.

| Dataset | #Seq. | Hierarchical Taxonomic Levels | | | | | | | Source | Tasks Type |
|---|---|---|---|---|---|---|---|---|---|---|
| | | Kindom | Phylum | Class | Order | Family | Genus | Specialized | | |
| Cirrhosis | 542 | 3 | 15 | 27 | 40 | 76 | 186 | 531 | Public Database | Binary-Classification |
| T2D | 606 | 3 | 17 | 29 | 48 | 94 | 216 | 587 | Public Database | Binary-Classification |

Cirrhosis and T2D. 'Kingdom', 'Phylum', 'Class', 'Order', 'Family', 'Genus', and 'Specialized' are all different taxonomic levels in the classification of organisms. Together, these taxonomic levels form the system of taxonomy, which is commonly used to describe and classify the planet's biodiversity. Each level represents a classification of organisms from broad to specific.

# D   Results

Appendix Table A5 and A6 show the M.F1 metrics for gene operon and pathogen prediction on two small-scale datasets, E-K12 and PATRIC (sequence length less than 5000).

# E   Implementation Details

In OpenMeta, we compare several genomic pre-trained models, including FGBERT, DNABERT2, NT, and HyenaDNA. Official implementations of these models can be accessed at the following URL links: HyenaDNA: https://huggingface.co/LongSafari/hyenadna-medium-450k-seqlen-hf, DNABERT2: https://huggingface.co/zhihan1996/DNABERT-2-117M, and Nucleotide Transformer: https://github.com/instadeepai/nucleotide-transformer. We have followed the default hyperparameters described in their respective publications and maintained consistent settings across all datasets, evaluating models at the checkpoints where validation loss was minimized. For sequence datasets, we investigate the impact of three encoding strategies on model performance: K-mer (K=3) frequency features, one-hot encoding features, and mean pooling embeddings from genomic and metagenomic models such as HyenaDNA, NT, DNABERT2, and FGBERT. The Macro F1-score (M.F1) is used as

Table A5: Gene Operon prediction on E-K12.

| Method | E-K12 |
|---|---|
| RF (3-mer) | 20.2 |
| SVM (3-mer) | 38.6 |
| AdaBoost (3-mer) | 39.9 |
| LSTM (w2v) | 40.4 |
| LSTM (one-hot) | 38.1 |
| BiLSTM (w2v) | 40 |
| BiLSTM (one-hot) | 40.1 |
| BiLSTM-Att. (w2v) | 38.2 |
| BiLSTM-Att. (one-hot) | 40.8 |
| VT | 43.3 |
| HyenaDNA | 42.4 |
| NT | 45.1 |
| DNABert2 | 51.7 |
| FGBERT | 61.8 |

Table A6: Pathegons prediction on PATRIC.

| Method | E-K12 |
| --- | --- |
| RF (3-mer) | 20.2 |
| SVM (3-mer) | 38.6 |
| AdaBoost (3-mer) | 39.9 |
| LSTM (w2v) | 40.4 |
| LSTM (one-hot) | 38.1 |
| BiLSTM (w2v) | 40 |
| BiLSTM (one-hot) | 40.1 |
| BiLSTM-Att. (w2v) | 38.2 |
| BiLSTM-Att. (one-hot) | 40.8 |
| VT | 43.3 |
| HyenaDNA | 42.4 |
| NT | 45.1 |
| DNABert2 | 51.7 |
| FGBERT | 61.8 |

the primary evaluation metric. In fine-grained sequence datasets, particularly the NCRD dataset for ARG prediction tasks, we evaluate three domain-specific models: the template-matching-based RGi, the deep learning-based DeepARG, and the pre-trained language model-based approaches PLM-ARG and FGBERT. Metrics used for evaluation included Accuracy, Precision, Recall, Macro F1-score, and False Negative Rate. For hierarchical datasets, due to the limited number of labels per hierarchical gene, we employed a variety of supervised models specifically designed for disease prediction, such as RF, SVM, Adaboost, and 1D-CNN, in addition to the specialized PopPhy model.

# F    Observations and Insights

**Fine-Grained Benchmarks.** Regarding ARG resistance category classification on the NCRD dataset, the metagenomic pre-trained method FGBERT outperforms the other three ARG prediction methods in all performance metrics and almost all resistance categories, as shown in Figure A1. The performance results in Table 10 show that DeepARG, a combination of traditional template matching methods and deep learning, performs well in the gene name and resistance categories but

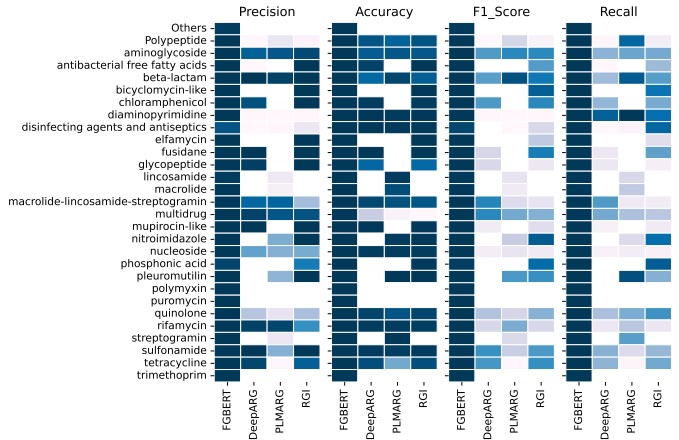

Figure A1: Performance comparison of FGBERT, DeepARG, PLM-ARG, and RGI on each antibiotic-resistant category.

fails to identify the gene family and mechanism categories, which is since no relevant data are included in the model training process or insufficient information is available in the matching dataset.RGI, as a template matching method, has a more general performance in all categories. PLM-ARG based on protein language modeling provided results and high false-negative rates only in the resistance category. FGBERT, as a metagenomic pre-trained model, performs well in all categories, demonstrating its comprehensiveness and high performance in dealing with fine-grained ARG assays, further proving the necessity and advantages of metagenomic pre-trained models.

## F.1   Genomic and Metagenomic Benchmark

**Relation between Genomic Benchmark and Metagenomic Benchmark.** While these benchmarks excel in single-species genome analysis, they often fail to capture the complex interactions among multiple species inherent in metagenomics. In contrast, metagenomic benchmarks aim to address interactions within diverse microbial communities and functional predictions in various environments,

covering tasks such as gene structure analysis, functional gene prediction, pathogenicity assessment, and nitrogen cycle prediction. Specifically, the limitations of genomic benchmarks include (1) their **reliance on single-species data**, which misses the complex interactions in metagenomics; (2) **lack of data diversity**, as their datasets are typically structured and uniform, lacking the environmental diversity needed for metagenomic studies; (3) **limited functional prediction**, focusing on sequence-based predictions without integrating crucial functional annotations; and (4) **insufficient model adaptability**, as models trained on single-species genomic data struggle with multi-species metagenomic data. These deficiencies underscore the urgent need to develop metagenomic benchmarks that can integrate multi-species interactions and complex environmental factors.

**(A) Metagenomic Pre-trained Models vs. Genomic Pre-trained Models:**

Table 14 compares genomic and metagenomic pre-trained models, including pre-training datasets, token embedding methods, network architectures, application tasks, and benchmarks. In terms of **Pre-Training Datasets**, DNABERT2 [84] utilizes human and multi-species genomes for its foundational model pre-training, covering a vast dataset of 27.5 billion nucleotide bases from the Human reference genome [32] and 135 species genomes across seven categories. HyenaDNA [49], on the other hand, is pre-trained solely on a single human reference genome. Nucleotide Transformer (NT) [9] pre-trains on three datasets: the human reference genome [32], 3,202 diverse human genomes, and 850 genomes from several species. In contrast, FGBERT [13] employs the MGnify database (updated February 2023) [62], comprising 2,973,257,435 metagenomic sequences from various microbial communities.

For **Token Embedding**, DNABERT2 applies Byte Pair Encoding (BPE) [67], while both HyenaDNA and NT use 6-mer tokenization. FGBERT utilizes a unique protein-based genomic representation tailored for metagenomic sequences.

Regarding **Network Architecture**, both DNABERT2 and FGBERT adopt BERT-like structures [11]; DNABERT2 enhances its predecessor by replacing learned positional embeddings with Attention with Linear Biases (ALiBi) [54] to eliminate input length limitations and incorporates Flash Attention [10] to boost computational efficiency. FGBERT introduces contrastive learning to strengthen the intricate relationships between metagenomic sequences and functions, proposing two pre-training tasks: Masked Gene Modeling (MGM) and Triplet Enhanced Metagenomic Contrastive Learning (TMC) to enhance co-representation learning of metagenomic gene sequences and functions. HyenaDNA employs a simple stack of Hyena operators for next token prediction, while NT uses an encoder-only Transformer architecture with Rotary Positional Embeddings (RoPE) [70] to enable reasoning over longer sequences during training.

For **Application and Benchmark**, Table 2 provides a detailed comparison of downstream tasks for genomic and metagenomic pre-trained models, highlighting the datasets, tasks, sequence numbers, and class counts. DNABERT2 focuses primarily on binary classifications of promoters and splice site detection tasks across human, mouse, yeast, and virus datasets, with the number in parentheses indicating the number of independent sub-datasets for each task and the sequence count reflecting the size of the first sub-dataset. Notably, **Transcription Factor Prediction** task recurs for both human and mouse species with identical dataset numbers, class numbers, and sequence lengths, suggesting a uniform level of difficulty that may not present significant challenges due to its repetitive nature across similar species settings. HyenaDNA's downstream tasks are divided into two parts: GenomicBenchmarks, consisting of 8 regulatory element classification datasets with sequence lengths ranging from 200 to 500, and NT's 18 prediction tasks. In addition to the human datasets, HyenaDNA includes Demo and Dummy datasets. The inclusion of **Demo and Dummy** datasets, which are typically used for initial testing and validation purposes. Additionally, NT covers 18 downstream tasks primarily centered around transcription factor binding, promoter prediction, and chromatin accessibility analysis, underscoring its detailed engagement with gene regulation mechanisms. FGBERT engages with various downstream tasks that address multi-species metagenomic sequences through multi-class classification challenges. These tasks span across gene, functional, bacterial, and environmental levels, accommodating input sizes that range from 1,000 to 219,000 sequences, reflecting the diversity and complexity of metagenomic data.

