# OpenReview forum: "OpenMeta: A Comprehensive Multi-Task Benchmark for Metagenomics Understanding"
_ICLR.cc/2025/Conference — Submitted to ICLR 2025_

### Official Review · Reviewer_AGLk · 2024-10-16

**Soundness:** 2
**Presentation:** 3
**Contribution:** 2
**Rating:** 3
**Confidence:** 5

**Summary:**

The paper presents a comprehensive multi-task benchmark. While the paper is well-structured and detailed, and shows it outperform other models in several tasks. There are several drawbacks. For example, although these models are often applied to various datasets, there are lack of interpretability for biologists. It also recognizes the need for a unified model that can handle both hierarchical and sequence data, but does not offer a concrete solution. Additionally, more evaluation metrics should be included, and real-world case studies would be added to strengthen the paper's practical value.

**Strengths:**

1. The paper provides a thorough comparison of different models, including both machine learning models and specialized genomic models.

2. The writing is well-structured and detailed.

**Weaknesses:**

1. The meta model can help raise a lot of hypothesis. It includes models like FGBERT and DNABERT2, but how interpretable are these models for biologists?

2. It seems like FGBERT can outperform other models on several tasks.  However, this pretrained model might be biased on the other tasks or datasets. The authors should discuss the limitations.

3.The paper mentioned the need for a unified model that can process both sequence and hierarchical data, but does not present concrete progress for addressing this gap.

4. The paper should include more evaluation matrix for fair comparisons.

5. There is no real-world validation, which should be included as the case study.

**Questions:**

What are the major technical challenges or limitations of OpenMeta, and how do you plan to overcome these in future development?

Please you give a few examples for real-world applications when applying this model.

---

### Official Review · Reviewer_74kr · 2024-10-22

**Soundness:** 2
**Presentation:** 2
**Contribution:** 3
**Rating:** 5
**Confidence:** 4

**Summary:**

The authors provide a benchmarking tool for metagenomic sequencing called openmeta. They motivate the development by the lack of appropriate existing tools and the significant difference between single organism sequence data (for which most tools are developed) and metagenomic data. The authors include more than 20 different approaches ranging from classic machine learning to pretrained large language model in their platform and evaluate them across a variety of datasets differing in task and complexity. The authors provide overarching conclusions from the benchmarking and in particular identify the need to also better include the hierarchical character of data.

**Strengths:**

+ The paper comprises a large variety of datasets and methods and thereby provides a broad overview
+ the paper succeeds in not only providing a benchmarking platform and evaluate, but also succeeds in reaching overarching conclusions how the field could move forward
+ the paper is uptodate with more recent developments in the fields and not just backwards looking

**Weaknesses:**

-	The paper overstates its uniqueness. The statement: “To our knowledge, there is no benchmark in the field of metagenomics.” is just incorrect. There are tons of benchmark, some of them, such as CAMI are community-run and cited >1000 times. Also for function prediction in metagenomics, there are tons of data sets. The uniqueness may be the step of putting it all into one place, running more current tools and making it available.
- The paper is surprisingly simplifying and often openly wrong in the understanding of metagenomics. For instance, Phylogenetics is commonly seen only as a pretty bad proxy of the actual taxonomy (which would be of interest here), e.g. https://www.nature.com/articles/s41587-020-0501-8. Metaphlan is a marker-gene driven approach, not a classic example of sequence alignment versus a reference.
The choice of negative data is absolutely crucial for many applications and seems to be rather ignored here (see e.g. https://academic.oup.com/bib/article/21/5/1596/5574719?login=false) and should be described in detail (and likely better included in the benchmark).
-	The frequent lack of articles and other grammatical mistakes make the manuscript unnecessary hard to follow.

**Questions:**

Why were these tools and datasets selected to form this benchmark. Can the authors prove more of a criteria-driven approach?

Can the authors clarify the novelty of their approach and contrast better to existent efforts?

How easily is openMeta extendable to novel developments?

How were negative groups defined for learning? E.g. for pathogenic potential?

**Details Of Ethics Concerns:**

The manuscript covers the task of pathogenicity prediction with large-language models. The authors touch these risks in their societal impact statement, but may not see the full scale of the problem (e.g. https://arxiv.org/pdf/2407.13059). While no novel methodology is provided per se, the authors evaluated models for this task and suggest improvements that have not been used before in this context.

---

### Official Review · Reviewer_tdNb · 2024-11-02

**Soundness:** 4
**Presentation:** 3
**Contribution:** 2
**Rating:** 5
**Confidence:** 4

**Summary:**

The authors present a benchmark for DNA models on metagenomics tasks. As I understand it, by "benchmark", the authors mean computing a set of evaluation metrics uniformly across a large number of models. The authors include many published DNA sequence models as well as vanilla versions of common ML models (random forest, LSTM, etc). The evaluation metrics the authors chose are the same as those used in Ref 13. Based on these evaluations, the authors come to several conclusions about the current state of the field.

**Strengths:**

Uniform benchmarking of computational methods is important to a healthy field. In particular, it is important include domain-specific benchmarks, as different models may be suited to different downstream tasks even if the mathematical form of their inputs and outputs are the same. The evaluation metrics the authors chose are reasonable. The manuscript contains a good review of challenges posed by metagenomics and discussion of the conclusions of their benchmark.

**Weaknesses:**

Impact/contribution: Overall novelty is low. Uniform benchmarking is important, but the most impactful papers include both a new model and the results benchmarking that led to its creation. The authors discuss the need for new methods, but stop short of developing them.

Presentation:
- The authors don't define a "benchmark" despite this being the core idea of the paper. Specifically, the text currently convolves: (1) an evaluation methodology; (2) the paper where this methodology was first proposed; (3) the implementation of this evaluation; (4) applying the evaluation to a specific model.
- It's hard to parse many tables of numbers. Using plots or at least bold/color would be helpful.
- The article has some grammatical errors, particularly regarding the usage of articles ("the", "a").

**Questions:**

Pending development of a new method, can one model be recommended above the rest for metagenomics tasks? This isn't explicitly stated.

---

### Official Review · Reviewer_8GJN · 2024-11-03

**Soundness:** 2
**Presentation:** 3
**Contribution:** 2
**Rating:** 1
**Confidence:** 5

**Summary:**

The paper proposed a metagenomic benchmark study with a construction of benchmark datasets and analysis of various machine learning methods.

**Strengths:**

Currently, pretrained/foundation models are more and more involved in metagenomics but extensive benchmark and standardized tests are still lacking.

**Weaknesses:**

1) The benchmark datasets seems to be merely pile-up of third-party datasets. The bio-science logic for selecting these datasets are not clear and weak. The catagory "small scale" and "large scale" are based on data size while the "fine grained" are based on problem definition and "hierarchical data" are based on pre-processing pipeline, which are not exclusive with each other. And it make no sense why Antibiotic resistence if "fine grained" than enzyme function?
2) The observations provided no valuable insights. Both points are common sense that already known for long time in microbiology.
3) The "Large scale" benchmark datasets are actually quite small for metagenomics. There are plenty of datasets with millions or even billions of sequences, and with much longer sequence lengths. For example, the NCBI virus database contained 3.4 million complete genome of annotated virus variants. HumanMetagenomeDB, TerrestrialMetagenomeDB, MarineMetagenomeDB, MetaGeneBank,..., all standardized and annotated and in size of TBs.
4) The two disease microbe datasets (tab. 5) were extracted from a single research paper each, which is unserious from  bio-science view.
There are existing disease microbe datasets (e.g. Disbiome) built by bioinformaticians and it took quite tedious procedures to reach concensus on confirming microbe-disease connections. For example, in Disbiome they collected over 10k experimental results to identify 1k microorganisms.

**Questions:**

N/A

---

### Meta-Review · Area_Chair_sDfF · 2024-12-19

**Metareview:**

This paper introduces a comprehensive benchmark, called OpenMeta, which is tailored for metagenomic function prediction.
The reviewers have raised multiple major concerns regarding the work, especially, regarding the overall value and significance of the proposed benchmarks.
The overall consensus is that the proposed benchmark does not add additional values beyond combining existing datasets, the benchmark size might not be sufficiently large for metagenomics prediction tasks, multiple simplifying and possibly erroneous statements are made, and there is significant room for improving the overall presentation of the work.

**Additional Comments On Reviewer Discussion:**

The authors have not responded to the reviewers' comments during the discussion period.
This leaves the reviewers' concerns unaddressed.

---

### Decision · Program_Chairs · 2025-01-22

Reject